# Rett Syndrome and the Role of MECP2: Signaling to Clinical Trials

**DOI:** 10.3390/brainsci14020120

**Published:** 2024-01-24

**Authors:** Adele Gaspar Lopes, Sampath Kumar Loganathan, Jayalakshmi Caliaperumal

**Affiliations:** 1Department of Pharmacology, Faculty of Medicine and Health Sciences, McGill University, Montreal, QC H3G 2M1, Canada; adele.lopes@mail.mcgill.ca; 2Cancer Research Program, Research Institute of the McGill University Health Centre, Montreal, QC H4A 3J1, Canada; sampath.loganathan@mcgill.ca; 3Department of Biochemistry, Faculty of Medicine and Health Sciences, McGill University, Montreal, QC H4A 3J1, Canada; 4Department of Otolaryngology, Head & Neck Surgery, Faculty of Medicine and Health Sciences, McGill University, Montreal, QC H4A 3J1, Canada; 5Departments of Experimental Surgery and Experimental Medicine, Faculty of Medicine and Health Sciences, McGill University, Montreal, QC H4A 3J1, Canada; 6Ingram School of Nursing, Faculty of Medicine and Health Sciences, McGill University, Montreal, QC H3A 2M7, Canada

**Keywords:** Rett syndrome, MECP2, neurodevelopmental disorders, trofinetide

## Abstract

Rett syndrome (RTT) is a neurological disorder that mostly affects females, with a frequency of 1 in 10,000 to 20,000 live birth cases. Symptoms include stereotyped hand movements; impaired learning, language, and communication skills; sudden loss of speech; reduced lifespan; retarded growth; disturbance of sleep and breathing; seizures; autism; and gait apraxia. Pneumonia is the most common cause of death for patients with Rett syndrome, with a survival rate of 77.8% at 25 years of age. Survival into the fifth decade is typical in Rett syndrome, and the leading cause of death is cardiorespiratory compromise. Rett syndrome progression has multiple stages; however, most phenotypes are associated with the nervous system and brain. In total, 95% of Rett syndrome cases are due to mutations in the *MECP2* gene, an X-linked gene that encodes for the methyl CpG binding protein, a regulator of gene expression. In this review, we summarize the recent developments in the field of Rett syndrome and therapeutics targeting MECP2.

## 1. Introduction

Rett syndrome was first identified by Andreas Rett in 1966. Rett identified a group of 22 girls who had similar medical histories, including clinical and neurological symptoms; indeed, he also noted how the mothers of the patients had higher rates of miscarriages and stillbirths before and after the patients’ births [1]. This was further elucidated by Fyfe et al., who saw double the amount of stillbirths in families who had Rett syndrome patients compared to controls, as well as more common perinatal losses on the maternal side [2]. The history and timeline of Rett syndrome, from identification to the major discoveries to date, is highlighted concisely in Figure 1.

Rett syndrome can be caused by mutations in the *MECP2* gene. The most frequent mutation location, reported by Bienvenu et al., is exon 3, which includes the methyl binding domain and the transcriptional repression domain of the gene [4]. Their group reported that the majority of mutation deletion events occurred in CCACC repeat sequences, which pointed to slippage mispairing. The other types of mutations, nonsense or frameshift, cannot produce proteins due to nonsense-mediated mRNA decay [4].

According to the Rett Syndrome (RTT) Variation Database [5], there are five point mutations that have a frequency of greater than 5% in Rett syndrome patients. This includes T158M, one of the most frequent missense mutations [5], which causes RTT through a mutant MECP2 that is unable to bind to DNA in experimental conditions [22]. The majority of point mutations are found in the methyl binding domains and the transcriptional repression domains of the *MECP2* gene. Missense mutations found in the C-terminal region and methyl binding domain of MECP2 disrupt the nuclear receptor co-repressor/silencing mediator of retinoic acid and thyroid hormone receptor (NCoR/SMRT) co-repressor interaction domain (NID) and destroy the interaction between MECP2 and this co-repressor complex. Thus, the neurological symptoms of RTT may be caused by the absence of MECP2 binding to NCoR/SMRT co-repressors and chromatin [6]. This review will highlight MECP2’s molecular functions and the promising basic and clinical research findings to date for Rett syndrome.

## 2. MECP2 Function

*MECP2* is found in the Xq28 long arm of the X chromosome. The gene itself is situated beside the interleukin-1-receptor-associated kinase gene and the red opsin gene that encodes a protein necessary for color vision [23]. The *MECP2* gene contains a 5′ untranslated region, four exons, and polyadenylated 3′ UTR. The MECP2 protein consists of repeated structures: the N-terminal domain and C-terminal domain as ends and a methyl CpG binding domain followed by an interdomain, AT-hook domain, interdomain, and a transcriptional repression domain, followed by another AT-hook domain and transcriptional repression domain. The transcriptional repression domain’s phosphorylation and activity depend on target gene activation [24].

Using NMR spectroscopy, the MECP2 protein was seen to adopt a wedge-shaped structure to undergo methyl binding flanked by arginine and lysine side chains on the DNA binding surface. There is no symmetry in the methyl binding domain [25]. The *MECP2* gene encodes for the methyl CpG binding protein that binds to CpG dinucleotides in the genome and was first identified to mediate repression through interacting with histone deacetylate and corepressor sin3A [7,8]. In contrast, when MECP2 works as a transcriptional activator, it uses CREB1 and a co-activator to bind to the genome [9]. Furthermore, MECP2 is the only protein from the methyl binding domain family that recognizes and selectively binds to CA repeats. It mediates neuronal function through the regulation of gene expression by modifying CA repeats [10]. In addition, MECP2 also interacts with Y box binding protein 1, which participates in the alternative splicing of mRNA and mRNA packaging and transport, as well as the repair and replication of DNA [26]. MECP2 was also seen to bind to histone deacetylase HDAC1/2 [7] and DNA methyltransferase DNMT1 [27], both controlling global gene expression (Figure 2).

The loss of MECP2 has, in some instances, led to an increased cholesterol level in *Mecp2* mutant mice and the general modulation of cholesterol homeostasis [28]. A dominant ENU mutagenesis suppressor screen showed that genes in cholesterol metabolism were upregulated and lipid metabolism was also disturbed. Here, the treatment of mice with statins to inhibit the first rate-limiting enzyme in cholesterol synthesis led to an enhanced median survival time and improved motor coordination and activity; however, other phenotypes like breathing irregularities were not improved with statin treatment [28]. Buchovecky et al. suggest that the changes in neuron maturation, synaptogenesis and the formation of dendritic complexes commonly seen in Rett syndrome patients could be explained by changes in cholesterol metabolism in the brain [28]. Furthermore, another study on plasma and cultured fibroblasts derived from Rett patients showed the modulation of proteins in the cholesterol network that could serve as therapeutic targets; a reduced level of 3-hydroxy-3methyl glutaryl coenzyme A (HMGR) activity that led to an increase in transcriptionally active sterol regulatory element binding proteins (SREBP-2); and rise in low-density lipoprotein receptor and HMGR protein content [29]. However, lovastatin was later shown not to improve motor performance and survival in *Mecp2* null mice as the treatment of statins was not necessarily followed by increased brain and serum cholesterol levels and an increase in motor performance and survival [30]. The genetic background could be a possible source of this discrepancy, suggesting that MECP2 deletion on its own does not affect brain cholesterol homeostasis and requires interactions with other genes. The group concludes that the efficacy of lovastatin is limited to RTT patients with alterations of cholesterol homeostasis and specific genetic backgrounds that are yet to be understood [30]. More recent work, published in 2022, involving a lipidomics analysis in the cerebrospinal fluid (CSF) and plasma of RTT patients, showed biochemical variations such as decreased cholesterol levels in CSF but not in plasma, and a decrease in phospholipid and sphingomyelin in CSF only [31]. These results point to the use of these proteins from patient CSF samples as RTT biomarkers and for improved lipidomics profiling.

The paradoxical nature of MECP2 to act as both a transcriptional activator and repressor led to the hypothesis that MECP2 has two isoforms generated through alternative splicing, which was later confirmed by Mnatazkanian et al. [32] and Kriaucinios and Bird [33] in 2004. The alternative splicing of the *MECP2* gene generates two isoforms of MECP2E1, with a translational start site at exon 1, and MECP2E2, with a translational start site at exon 2, which have shared exons 3 and 4, where the majority of RTT-causing mutations occur. However, MECP2E1 is the predominant isoform found in the brain [34]. Although both isoforms have overlapping and different functions, researchers have shown that MECP2E2 is a player in ribosomal gene expression regulation, while MECP2E1 is more involved in neuronal chromatin structure and gene expression control [35].

## 3. MECP2 Variants

The pathophysiology from mutation to disease symptoms remains unclear. However, there has been recent work attempting to correlate the mutation of *MECP2* with the clinical severity of the disease. Patients with pathogenic truncating mutations of a nonsense or frameshift nature had a more severe phenotype than those with missense mutations. Furthermore, there was no difference in severity between missense mutations found in the methyl binding domain compared to those in the transcriptional repression domain [36]. This group also found that 69% of unrelated RTT cases caused by *MECP2* mutations showed C to T transitions in CpG dinucleotides that pointed to the CT doublet’s hypermutability. This offers a potential drug target for pregnant mothers if a drug can be found to stabilize the doublet in this genome during development. *MECP2* mutations are not only found in girls but also have lately been found in boys, producing an atypical RTT phenotype and other less distinctive RTT features that has led to a new diagnosis of male RTT encephalopathy [37].

The *MECP2* gene is found on the X chromosome, and, since women have two copies, one must be inactivated in the cell [38]. Random X inactivation ensures the correct number of gene dosages in each cell in a woman’s body and provides dosage compensation between sexes [39]. The inactivation of the X chromosome in each cell is randomly chosen and initiated by the noncoding Xist RNA that covers the inactive X chromosome; females thus have a mosaic pattern of active and inactive X chromosomes throughout the cells in their body [40]. Non-random or skewed X inactivation involves an X chromosome with mutant alleles, with an increased possibility of being inactivated compared to the wild-type gene, and this is common in X-linked neurological disorders [11,41]. Thus, phenotypic variability arises from skewed X inactivation. Researchers have found, in patients with classical RTT, the preferential inactivation of the paternal X chromosome, and the paternal X chromosome frequently carries de novo *MECP2* mutations [42]. Furthermore, Fang et al. showed that common RTT mutations led to higher levels of skewed X inactivation and predicted that increasing skewed X inactivation could cause more severe phenotypes. Multiple studies have shown the relationship between the mutation type and clinical severity for Rett patients. P.A133C, p.A306C and p.A306C, as well as truncations and other point mutations, were less severe in typical and atypical RTT, while p.A106T, p.A168X, p.A255X, p.A270X and R168X, as well as deletions and insertions, were found to create more severe clinical presentations [43,44].

The MECP2 protein levels must be maintained tightly because underexpression leads to Rett’s syndrome, while overexpression leads to MECP2 duplication syndrome. In neurons, particularly, the doubling of MECP2 levels leads to an increased synaptic response, while hippocampal glutamatergic neurons that do not express MECP2 show a 46% decrease in synaptic response [12]. These in vivo changes in synapse number and response were noted in early post-natal development and disappeared over time, suggesting a homeostatic physiological response over time. These in vivo changes could also reflect the change in phenotype progression for Rett syndrome patients. Chao also suggests that MECP2 regulates genes required for the initial formation of synaptic contacts [12].

MECP2 is a newly identified subunit of the Rbfox/LASR complex that regulates splicing and post-transcriptional regulation in the brain [13,14]. It was further elucidated that MECP2 interacts with RBFOX2, another protein involved in alternative exon splicing in the nervous system, through its methyl binding domain and interdomain. Furthermore, Jiang et al. [13] also applied the frequent Rett-syndrome-causing T158M point mutation to mice and revealed the inability of MECP2 to act as a platform to create the Rbfox/LASR complex in cultured cells and the mouse model brain, thus causing erroneous splicing. The following year, researchers also found that the majority of RTT missense mutations destroyed MECP2’s interaction with the nuclear receptor corepressor (NCoR) and silencing mediator of retinoic acid and thyroid receptors (SMRT) corepressor complex [45]. The structure of the contact between MECP2 and the NCoR/SMRT interaction domain (NID) showed that the missense mutations caused interacting residues of transducing beta-like 1 (TBLR1) and transducing beta-like related 1 (TBL1) to disrupt the binding of MECP2. The mutated residues in RTT that display the most contact with TBLR1 are associated with intellectual disability [45].

Other diseases related to Rett syndrome are MECP2 duplication syndrome, FOXG1 syndrome and CDKL5 deficiency disorder, which are other single-gene neurodevelopmental disorders; all four of these disorders result in the reduced amplitude of visual evoked potentials, which could serve as biomarkers for the disease [46]. Furthermore, the more reduced the amplitude is, the greater the clinical severity for RTT patients reported.

MECP2 duplication syndrome symptoms include a lack of communication, abnormal walking and balance issues, constipation and seizures. Clinical severity has been noticed to increase with older age because of increasing motor dysfunction [47].

## 4. Stages of RETT Syndrome and Symptoms

Currently, Rett syndrome is diagnosed through clinical symptoms because *MECP2* mutations do not necessarily cause symptoms due to the excessive X chromosome inactivation on the loci with the mutation. Furthermore, *MECP2* mutations cannot be found in some patients presenting with classical RTT [9].

Because of the similarity between autism and Rett syndrome, clinicians often misdiagnose Rett syndrome as autism in young girls; both conditions present social and learning development difficulties and stereotyped hand movements. In a survey of RTT cases from Australia, the USA and the UK, 17.6 % of cases had an early diagnosis of autism, with the two mutations of p.T158M and p.R255X (milder) most likely to result in the initial misdiagnosis of autism. Furthermore, the likelihood of a misdiagnosis of autism increased by 1.02 for every increase in the months in which a loss of hand function and loss of communication was reported [48].

Rett syndrome is generally categorized into typical RTT and variant forms of RTT. In a recent update to the nomenclature for RTT, researchers have established revised diagnostic criteria. The symptoms required for typical RTT include a period of regression followed by recovery or stabilization, as well as all main and exclusion criteria [15]. The main criteria include (i) the partial or complete loss of acquired purposeful hand skills, (ii) the partial or complete loss of acquired spoken language, (iii) gait abnormalities that are impaired or an absence of ability and (iv) stereotypic hand movements such as squeezing, clapping and rubbing movements [15]. Some supportive criteria include breathing disturbances when awake, impaired sleep patterns and abnormal muscle tone. Contrasting typical RTT phenotypes are the three variant forms of RTT: the preserved speech variant (Zapella) [49], the early seizure variant (Hanefield) and the congenital variant (Rolando). The Zapella variant is the only variant where *MECP2* mutations are mostly found, while both the Hanefield and Rolando variants rarely contain *MECP2* mutations and should be screened for mutations in CDKL5 and FOXG1, respectively [16,17]. Individuals with RTT also have sleep problems, such as sleep-disordered breathing, and report higher daytime sleepiness [50].

The onset of typical disorder symptoms starts after the age of 6 months, where neuronal development has already produced severe changes [51]. Up until the age of 5 years old, children will suffer from feeding difficulties and low levels of growth, as well as gastrointestinal issues and head growth stagnation [52]. In late childhood, before puberty, the regression phase ends [53] and stabilization starts (Figure 3). The stagnation stage can be anywhere between 2 and 10 years in length and may include recovery and gains in mobility; however, patients still have intellectual disability, seizures and gastrointestinal problems [51]. An analysis of North American Rett syndrome patients also shows that more Rett syndrome patients are surviving to middle age, with a 50% chance of survival at 55 years old [54].

Only in the past decade have we begun to uncover the full extent of the Rett syndrome phenotype and, thus, the potential actions of MECP2 as a gene regulator. First, there is a link between thyroid function and Rett syndrome. Patients with *MECP2* mutations and deletions showed higher levels of thyroid hormone FT_4_ than controls, and half of this group showed more elevated thyroid hormone FT_3_ levels than controls [55]. This research suggests that more analyses should be performed regarding the link between Rett syndrome and thyroid function; specifically, patients should be screened for potential thyroid dysfunction and treated with relevant thyroid drugs. Second, in comparison with the general female population, more than 25% of RTT patients initiated puberty early, but 19% experienced a delayed menarche; the relationship between the mutation type and puberty trajectory remains unclear [56]. Furthermore, Rett syndrome patients’ anthropometric measurements varied depending on the severity of their RTT phenotype; those with classic RTT had increased arm, thigh and lower leg muscle measurements over time, while mild and severe atypical RTT patients did not follow this pattern. This suggests that these types of measurements can be used to provide evidence of effectiveness in clinical trials [52].

When patients present with Rett syndrome, parents are asked to complete the Rett Assessment Rating Scale, which evaluates the patient’s phenotype to classify its severity [57]. The seven subgroups of the Rating Scale include sensory, cognitive, motor–structural, functional, emotional, autonomy, physical and behavioral characteristics. After evaluation on a 7-point ordinal scale, scores from 0 to 55 indicate mild impairment, 58–81 indicate moderate impairment and 82–124 indicate severe impairment and clinical severity [58]. Other assessments include the clinical severity scale utilizing Likert measures with scores from 0 to 58 and the Motor Behavior Assessment on an ordinal score from 0 to 4, with 0 being the worst and 4 being the best. This measures behavior and socializing, respiratory health and physical movement. Better results on the MBA were found to be correlated with worse psychological severity and vice versa [59]. The assessment score also increases with age and is strongly correlated with clinical severity assessed using the RTT clinical severity scale and the expected relationship between the genotype and phenotype [60]. This suggests that the MBA score could be used to measure clinical trial success for future treatments. Another assessment created by researchers and physicians is called the Caregiver Assessment of Symptom Severity (RCASS), used to assess caregiver-reported outcomes [61]. The RCASS is the first to combine the following four factors: movement, communication, behavior and Rett-specific symptoms. These are also the primary concerns indicated by caregivers or participants [62]. Criterion validity has been established with the Revised Motor Behavior Assessment Scale, Clinical Severity Scale, Clinical Global Impression Scale and Child Health Questionnaire. These assessments for severity may now be used in clinical trials in RTT to measure the outcomes of the disease [61].

## 5. Signaling Events Tied to RTT Disease Progression

It is difficult to pinpoint exactly which biological pathways are affected by a mutant MECP2 to create the different stages of Rett syndrome, because timing of the disease cannot be easily replicated in vitro, and the disease is caused by a multitude of different mutations. These different mutations have been shown to affect different sets of genes; thus, a different set of physiological pathways will create a phenotype of Rett syndrome that is unique to each patient [63].

MECP2 has been shown to globally impair immediate early genes (IEGs), which means that MECP2 binds to and regulates IEGs that are needed for correct synaptic development [18]. This is contrary to previous work conducted on MECP2 that concluded that it mediates the final stages of neuronal development, such as axonal projections and axonal targeting in the olfactory bulb [64], neuron maturation and synaptic maintenance [65]. Petazzi and their team found high levels of expression of IEGs in RTT patients’ post-mortem tissue, which showed the dysregulation of these sets of genes; indeed, four IEGs, FOS, JUNB, EGR2 and NPAS4, showed altered expression in MECP2-KO cultured neurons. Thus, it remains unclear why RTT patients do not show symptoms earlier in their lifetime, but these findings elucidate the malfunctioning pathway of synaptic plasticity contributing to the poor cognitive, learning and memory skills of RTT patients [18]. Furthermore, the dysfunction of IEGs will impact their transcriptional targets, such as genes involved in the myelination process. However, more work needs to be done to elucidate the specific alterations taking place in the myelination pathways downstream of a *MECP2* mutation.

Furthermore, MECP2 is associated with the centrosome and is involved in the formation and functioning of the primary cilium [19]. The primary cilium is composed of microtubules and grows from a centrosome-derived structure [66]. By identifying primary cilium dysfunction in vitro with cells from RTT patients, Frasca et al. found the functional impairment of the ciliary-related Sonic Hedgehog signaling pathway, thus showing that the selective inhibition of HDAC6 to stabilize alpha-tubulin recovers phenotypes and it may act as a pharmacological target for RTT. Finally, researchers have established that MECP2-null cerebella show reduced levels of Gli1, a zinc finger protein involved in invasion and transition in glioma cells, and other members of the same pathways [19].

Recently, more studies have examined the role of the gut microbiota in the progression and symptoms of Rett syndrome. Firstly, Strati et al. showed that Rett patients had an altered bacterial and fungal microbiota and reduced microbial diversity that did not depend on the constipation status of the patient [67]. A proinflammatory status of the gut microbiota is hypothesized as a decrease in *Prevotella* and *Faecalibacterium* has been shown, as well as a decrease in Bacteroides and increase in actinobacteria [68]. Furthermore, the characterization of the Rett patient gut microbiota and examination of short-chain fatty acid (SCFA) concentrations showed lower alpha diversity and increased concentrations of branched SCFAs in Rett patients, suggesting an increased number of microbial genes encoding for amino acid metabolism [69]. This has led to questions about the correlation between the neural symptoms of Rett syndrome and microbiome alterations. In female rat models of RTT, neural symptoms occurred before significant gut microbiome alterations, but motor abnormalities appeared after the changes in the gut microbiome, which suggests a role of microbial changes impacting the severity of motor symptoms in Rett syndrome [70]. 

Contrary to previous findings, Neier et al. found no significant differences in microbiota diversity in their female Rett mouse model in comparison to wild-type males, but they did report altered gut microbial communities and inflammatory profiles [71]. Interestingly, fecal metabolites were altered in *Mecp2e1* mutant females before the onset of neuromotor phenotypes and were correlated with lipid deficiencies in the brain. Through weighted gene co-expression network analysis, fecal metabolites were shown to be sensitive indicators of RTT progression in females. Lipidome alterations such as downregulated phosphatidylethanolamines and sphingomyelins in the cortex correlated with fecal metabolic changes. Further, there was a high degree of inverse correlation between the top 10 genotype-associated fecal lipids at 9 weeks of age and the top 10 genotype-associated cortical lipids at 19 weeks of age in females; in particular, the lipids increased in fecal matter were associated with decreased lipids in brain cortices. These decreased lipids are critical for neuronal function, suggesting that the decreased absorption of lipids in the GI tract may negatively impact brain lipids in females with RTT [71]. The examination of microbiome and metabolome modulations, and the changes in absorption of lipids, may help us to better understand the pathogenesis of Rett syndrome and thus help with understanding the mechanisms underlying the symptoms of this disease.

## 6. Current Clinical Treatments and Drugs

Rett syndrome is a challenging disease to treat because of its multi-system nature and the many unknown pathways that are involved in the disease. Here, we will review the current clinical treatment of RTT and other technologies used to assess the disease, as well as the current clinical trials underway to treat RTT (Figure 4). In 2022, the first peer-reviewed consensus-based therapy guideline was published. The combined clinical trials centered around patients and families, and the increasing medical knowledge emerging on the molecular details of RTT, mean that individuals with RTT are surviving into adulthood. However, guidance is needed to outline considerations for healthcare professionals and parents with RTT children as it is important to integrate the care provided for this multi-system disease [62]. Fu et al. created a framework for baseline, annual and biannual treatments for patients with RTT. These include, but are not limited to, screening for awake disordered breathing and air swallowing, screening for the presence of seizures or non-epileptic spells, 12-lead ECGs and estimating the curvature of the spine.

In terms of treatment for patients suffering from RTT, the past two decades have seen a surge in the usage of wearable technologies as therapeutic aids for patients. Eye gaze technology is expanding the world of communication and learning for those with RTT [72]; eye gaze technology was also used in the Netherlands as a form of augmentative and alternative communication for individuals with RTT, in an attempt to solve the apraxia of speech problem, since patients understand more than they can express themselves [73,74]. The children created illustrations using photos and graphic symbols, and parents reported an increased understanding of language. Wearable sensor technology has been used in RTT patients to assess emotional, behavioral and autonomic dysregulation through electrodermal activity and heart rate variability [75]. Wearable sensors were used to monitor patients after treatment with buspirone, sertraline and gabapentin; three quarters of patients treated with buspirone saw improvements in EBAD, while both treated with sertraline saw improvements.

Further, wearable sensors have been used for the therapeutic monitoring of autonomic dysregulation in RTT. Assessing heart rate variability allows researchers to examine the day and night changes in patients and frequency domain sympathetic and parasympathetic indices. One study saw that RTT patients were less adaptable to autonomic changes at night and the heart rate decreased with age and was lower at night [76]. The usage of telerehabilitation systems has also been shown to be feasible for RTT patients to increase joints’ passive range of movement in the upper body in a recent pilot study [77].

There have also been emerging therapies for the treatment of RTT in both clinical trials and mouse models, and some of the completed trials are listed in Table 1 and Table 2. Firstly, Smith et al. used transcriptional profiling in autopsy samples from patients to identify M1 muscarinic acetylcholine receptor potentiation as a target for therapy [78]. M1 expression and MECP2 expression had a linear relationship in this study. M1 potentiation rescued social preference, spatial memory and memory deficits, as well as decreasing apnea, in *Mecp2* hetero (+/−) mice, and it normalized global gene expression patterns. There is potential for the selective targeting of M1 receptors with positive allosteric modulators that lack agonist activity [79,80]. Neurological defects have also been reversed in a mouse model of Rett syndrome, showing that the viable mutant neurons present in the brain can be repaired by the activation of the MECP2 gene and expression; mice reverted to a phenotype that was close to the wild type in terms of inertia, irregular breathing, abnormal gait and hindlimb clasping [81]. Further, long-term potentiation in the hippocampus was restored with MECP2 expression, showing that, due to a lack of permanent neuronal cell death, MECP2 only reversibly damages neurons, suggesting that RTT is not a neurodevelopmental disorder [81].

Researchers have also established a method to reactivate the inactive X-linked *MECP2* in the cerebral cortical neurons of living mice to correct symptoms of defective neurons, such as soma size and branch points [105]. This group used pharmacological inhibitors targeting X chromosome inactivation (XCI) promoting factors to reactivate X-linked *MECP2* in cultured mouse fibroblasts, human induced pluripotent stem cells from RTT patients and the cerebral cortical neurons of living mice. The inhibitors decrease Xist expression by preventing the recruitment of transcription activators to the X-inactive specific transcript (Xist) promoter and offer an excellent drug target because they directly provide more MECP2 instead of affecting a pathway downstream. The mechanistic target of rapamycin (mTOR) inhibitors that were used in this study are also currently being used in therapeutics for cancer and neurodegenerative diseases [106,107,108].

As mentioned previously, there is ongoing research involved in understanding the relationship between the gut–brain axis, the microbiome and Rett syndrome disease symptoms and progression. Treating Rett female mice with 75 mg/kg per day of Leriglitazone, a peroxisome proliferator-activated receptor gamma (PPARy) agonist that inhibits inflammation and oxidative stress, led to the recovery of bioenergetic alterations in both Rett fibroblasts and mice and an anti-inflammatory effect on Rett mice. These bioenergetic alterations were most significant in the cerebellum in the Rett mice models [109].

Building on the usage of the *MECP2* gene, researchers have also shown that MECP2 supplementation in the R249X mouse model of RTT rescues behavioral phenotypes in both male and female mice. However, the female mice dosage for supplementation must be carefully monitored to avoid overdosage and, thus, adverse motor effects and changes in motor circuitry [110]. The treatment of female heterozygous mice with recombinant human insulin growth factor 1 showed improvements in physiological and behavioral symptoms as well as improvements in the maturation of cortical circuits [111]. The first drug with evidence supporting its usage as a treatment for RTT patients is fluoxetine. In a mouse model of heterozygous *Mecp2* mice, fluoxetine increased MECP2 expression in the brain by increasing the number of MECP2+ (immuno-positive cells) in the prefrontal cortex and motor cortices. The usage of fluoxetine to induce an increase in MECP2 could potentially rescue motor coordination [20].

The second drug recently approved by the FDA to treat RTT is trofinetide [112]. In a double-blind, randomized, placebo-controlled clinical trial named LAVENDER, a phase 3 study of trofinetide [21] evaluated 187 females with RTT and showed improvements for patients treated with 50, 100 or 200 mg/kg received twice a day [113,114]. Specifically, trofinetide at 200 mg/kg showed the most significant and clinical improvement relative to the placebo, as evaluated through the Rett syndrome behavior questionnaire, RTT clinical domain-specific concerns including the visual analog scale and clinical global impression scale improvements [113]. Trofinetide is a synthetic analog of glycine-proline-glutamate, a naturally occurring tripeptide in the brain that is cleaved from insulin-like growth factor 1 [62]. Trofinetide was shown to increase symptoms associated with RTT for the 75% of participants receiving trofinetide who completed the study; however, there were side effects of mild or moderate diarrhea and vomiting. The research team implemented a diarrhea management plan partway through the study to adjust for trofinetide, since RTT patients commonly take laxatives for their GI issues [62]. Trofinetide is also continuing to be studied pharmacokinetically and analyzed for its safety in RTT patients younger than the age of 5 and participants may roll over into an open-label LILAC study [115] or a LILAC extension study [116] to examine the long-term effects and safety of using trofinetide to treat RTT. The molecular mechanism of trofinetide remains unclear.

There are also current gene therapy clinical trials underway targeting *MECP2* dosage levels. Firstly, an active phase 1/2 clinical trial [117] is assessing the efficacy of administering adeno-associated viral vector serotype 9 (AAV9) using transgene regulation technology. Neurogene Inc. will be administering a full-length human MECP2 gene designed to express therapeutic levels of MECP2 via intracerebroventricular delivery. Taysha Gene Therapies Inc. is also overseeing a phase 1/2 study of TSHA-102, an AAV9 gene transfer therapy that uses miRARE to regulate *MECP2* expression. Finally, another group of researchers is investigating the use of CRISPR/cas9-based gene editing with AAV delivery to correct *MECP2* mutations in vitro and in vivo [118].

## 7. Conclusions: Lessons Learned and Looking Ahead to Rett Treatment

In conclusion, there have been major improvements in the past 10 years in terms of understanding the mechanism of the MECP2 pathways in the body as well as how the MECP2 mutant contributes to Rett pathology. This year has seen a strong candidate, trofinetide, enter the market for the treatment of Rett, and the upcoming clinical trials mentioned and newly revised diagnostic criteria for patient care guidelines make the treatment of Rett patients promising for clinicians, families and personal care workers. Looking ahead, more work needs to be conducted to understand the molecular reasons for the change in disease stage progression in Rett syndrome patients. Furthermore, more needs to be understood to elucidate the role of MECP2 in the brain and across the body. Once we understand the MECP2 mechanism, we can continue working on gene therapy with *MECP2* or suitable downstream targets.

## Figures and Tables

**Figure 1 brainsci-14-00120-f001:**
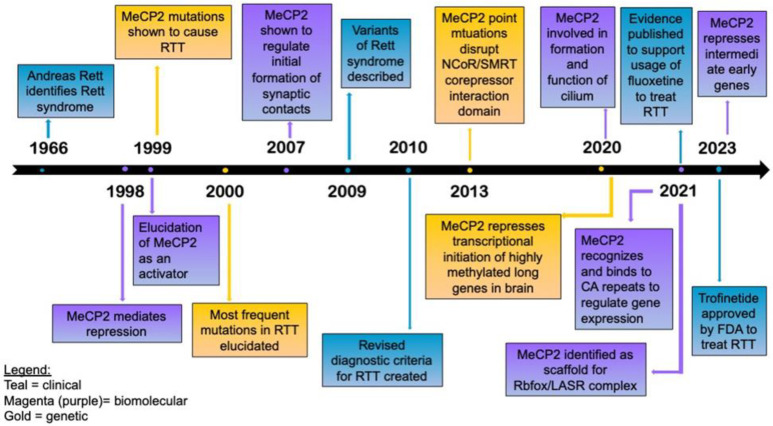
Timeline of Rett syndrome, from identification to the current clinical trials and treatment. During 1998–99, mutations in *MECP2* were linked to Rett syndrome. Throughout the subsequent decades, the molecular function of MECP2 was further elucidated. In 2023, the first FDA-approved clinical drug for the treatment of Rett syndrome entered the market in the USA [1,3,4,5,6,7,8,9,10,11,12,13,14,15,16,17,18,19,20,21].

**Figure 2 brainsci-14-00120-f002:**
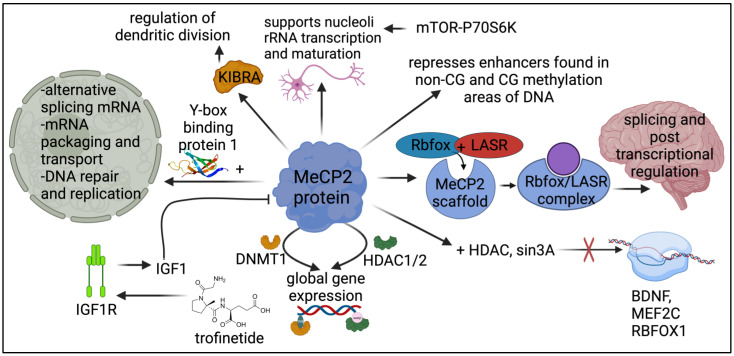
Overview of MECP2 protein activity pathways. MECP2 has a role in over ten different pathways, forming complexes such as (from left to right, clockwise) the Y box binding complex, the Rbfox/LASR binding complex and the MECP2-HDAC-sin3A, MECP2-HDAC1/2 and MECP2-DNMT1 complexes. Further, the protein plays a large role in gene expression, transcription regulation and neuronal development. The specific modulation of IGF1 on MECP2 remains unclear.

**Figure 3 brainsci-14-00120-f003:**
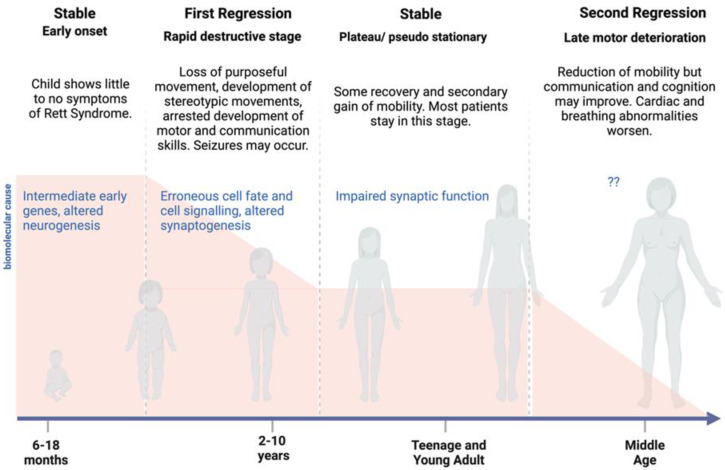
Disease progression stage in Rett syndrome (Kirby, 2010 [54]). There are multiple possible molecular causes of the disease at an early stage and at first regression depending on the phenotypic progression at this point. The molecular cause for the second regression stage remains unclear; however, only a proportion of Rett patients survive to reach the second regression stage. Currently, the pathogenesis of Rett syndrome is better understood in terms of earlier stages of the disease than in the teenage and young adult years.

**Figure 4 brainsci-14-00120-f004:**
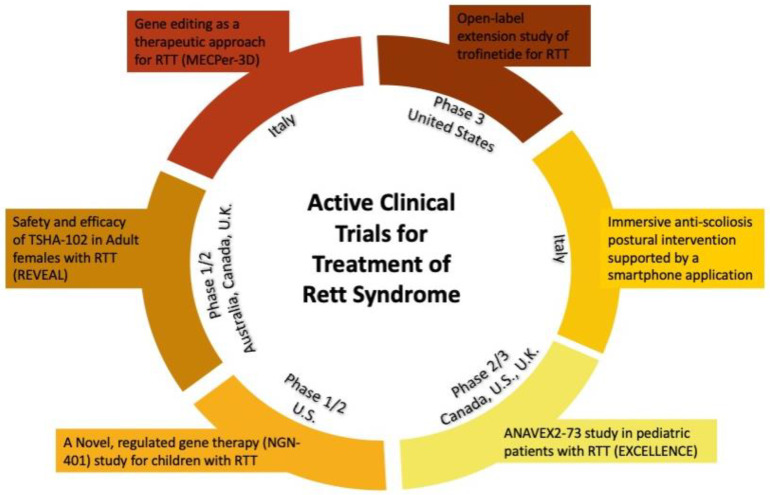
Summary of current active clinical trials for treatment of Rett syndrome. There is large diversity in location; however, there are two gene-focused clinical trials, gene therapy and gene editing. Furthermore, trofinetide has shown great promise from phase 2 results and has moved on to phase 3 clinical trials. Finally, in Italy, a clinical trial is focused on improving posture through extra support for caregivers and patients.

**Table 1 brainsci-14-00120-t001:** List of completed clinical trials for RTT with results.

Clinical Trial	Date	Location	Responsible Company	Description	Results
Pharmacological treatment of RTT with glatiramer acetate (Copaxone) NCT02153723 [82]	4 October 2018 and updated 5 November 2018	Bronx, New York, United States	Montefiore Medical Center	Phase 2 open-label trial to test Copaxone (glatiramer acetate), which is normally used in treating multiple sclerosis. An increase in BDNF levels leads to delayed onset of Rett symptoms, and Copaxone treatment causes elevation of BDNF. Expected to see increase in BDNF levels with Copaxone administration via subcutaneous injection, which will lead to amelioration of symptoms.	1/10 participants experienced musculoskeletal and connective tissue disorders with elevated CPK. Gait velocity was improved, memory and breath holding index also improved. Epileptiform discharges decreased in four patients who had the same value as baseline. Trial arrested due to three of 14 patients developing life-threatening postinjection response.
Placebo-controlled trial of dextromethorphan in RTT NCT01520363 [83]	7 November 2018	Baltimore, Maryland, United States	Hugo W. Roser Research Institute at Kennedy Krieger Inc. (John Hopkins), Baltimore, USA	NMDA glutamate receptors are increased in the brains of RTT children less than 10 years old, and their excess causes over-stimulation of nerve cells. Dextromethorphan was proposed because it has been shown to block NMDA receptors.	No serious adverse effects were reported in 26 patients treated with DM in comparison to the placebo group. Non-serious adverse effects that were reported were increased platelets and increased alkaline phosphatase.Improvements were seen in clinical seizures, repetitive language and behavioral hyperactivity. No significant increase in global clinical severity scale as measured by RTT severity scale.
Trial of dextromethorphan in RTT NCT00593957 [84]	23 April 2014	Baltimore, Maryland, United States	Hugo W. Roser Research Institute at Kennedy Krieger Inc. (John Hopkins), Baltimore, USA	NMDA glutamate receptors are increased in the brains of RTT children less than 10 years old, and their excess causes over-stimulation of nerve cells. Dextromethorphan was proposed because it has been shown to block NMDA receptors.	Trial terminated because Food and Drug Administration required placebo-controlled trial instead of ongoing open-label trial and recruitment was delayed.
Treatment of RTT with recombinant human IGF-1 NCT01777542 [85]	26 March 2018	Boston Children’s Hospital	Harvard Medical School	Double-blind study using recombinant human IGF-1 (mecasermin or INCRELEX), an investigational drug approved for children by the FDA, to test for improvement in symptoms of RTT.	Patients treated with IGF-1 had increased risk of colitis, but it also improvements in some breathing behaviors and behavioral abnormalities. Moreover, anxiety and mood were noted to improve.
Pharmacological treatment of RTT with statins NCT02563860 [86]	12 July 2019	Bronx, New York, United States	Aleksandra Djukic, Montefiore Medical Center	Phase 2 open-label dose escalating study of Lovastatin to inhibit cholesterol synthesis in the CNS and reduce high neuronal cholesterol in RTT patients.	Treatment was seen to improve visual recognition, memory and eye tracking through the Rose neuropsychological test. Only 1/20 patients reported non-serious adverse effects of mild psychiatric disorders.
Evaluation of the efficacy, safety, and tolerability of Sarizotan in RTT with respiratory symptoms NCT02790034 [87]	21 December 2021	United States, Australia, India, Italy, United Kingdom	Newron Pharmaceuticals SPA	Randomized, double-blind, placebo-controlled study to assess efficacy of multiple doses of Sarizotan in patients with RTT focusing on improving respiratory health.	This study was terminated during double-blind period because it did not demonstrate evidence of efficacy on variables of reduction in respiratory abnormality in patients and assessment by caregiver-rated impression of change.
A long-term safety study of cannabidiol oral solution in patients with RTT NCT0425286 [88]	8 August 2022	United States, Australia, Canada, Italy, United Kingdom	Jazz Pharmaceuticals	Long-term safety of cannabidiol oral solution in participants with RTT.	43% of patients had seizure frequency reduced by half, and 5% became seizure-free. However, patients suffered from serious adverse side effects such as diarrhea, vomiting, fatigue, pyrexia and somnolence. Study was terminated due to COVID-19 pandemic and recruitment challenges.

Summarized clinical trials. CPK is creatine phosphokinase, BDNF is brain-derived neurotrophic factor, NMDA is N-methyl-D-aspartate, DM is dextromethorphan, human IGF-1 is insulin growth factor 1, CNS is central nervous system.

**Table 2 brainsci-14-00120-t002:** Completed/terminated clinical trials with results not linked on clinicaltrials.gov website.

Clinical Trial	Location	Date	Responsible Party	Description	Results
Study to assess safety and efficacy of fingolimod in children with RTT (FINGORETT) NCT02061137 [89]	Basel, Switzerland	15 June 2018	University Hospital, Basel, Switzerland	Phase 1 clinical trial to assess safety and efficacy of oral fingolimod (FTY720) in children older than 6 years old with RTT. FTY720 increases levels of brain-derived neurotrophic factor (BDNF), which is low in RTT patients.	No serious adverse events occurred but changes in BDNF levels in CSF and serum, deep gray matter volumes and clinical scores were not met. No supportive evidence to use fingolimod to treat RTT.
Tolerability of immersive virtual reality system GRAIL in subjects affected by RTT NCT05691582 [90]	Bosisio Praini, Lecco, Italy	20 January 2023	IRCCS Eugenio Medea	Investigating tolerability of GRAIL system in subjects affected by RTT. Aimed to see if GRAIL, a virtual reality system on a treadmill, could improve gait characteristics in RTT.	No results posted.
A safety study of NNZ-2566 (trofinetide) in RTT NCT02715115 [91]	Birmingham, Oakland, San Diego, Aurora, United States	14 August 2020	Neuren Pharmaceuticals Limited	Phase 2, double-blind, placebo-controlled study to investigate whether NNz-2566 (trofinetide) is safe and tolerable treatment for oral administration in children and adolescent females with RTT.	Safety and tolerability of trofinetide was very good, with mild side effects being diarrhea, vomiting and pyrexia. Trofinetide showed significant clinical improvement compared to the placebo group.
A safety study of NNZ-2566 (trofinetide) in RTT NCT01703533 [92]	Birmingham, Saint Paul, Houston, United States	5 February 2018	Neuren Pharmaceuticals Limited	Phase 1 of the same study.	Phase 1 completed and approved.
Osteopathic manipulative treatment for constipation in people with RTT NCT05687214 [93]	Verona, Italy	1 February 2023	Ariel University	Constipation negatively impacts quality of life mentally and physically. Single-blind study to evaluate efficacy of OMT for management of chronic constipation.	No results posted.
Development of the ORCA communication measure for RTT NCT04920110 [94]	Durham, United States	22 April 2022	Rett Syndrome Research Trust	The Observer-Reported Communication Ability (ORCA) is a 72-question questionnaire that detects change over time in an individual’s types of expressive, receptive and pragmatic communication. ORCA allows caregivers to assess communication of the patients independently. This phase 2 study will assess whether ORCA is valid for RTT patients.	Data contributed by parents are being analyzed. ORCA will be available to the biopharmaceutical community to test in clinical settings.
Assessing Emerald and MC10 BIOSTAMP nPOINT BIOSENSORS for RTT NCT04514549 [95]	Boston, United States	28 March 2023	Rett Syndrome Research Trust	Pilot study of Emerald device to monitor sleep, breathing and movement and nPoint patches to determine proper patch placement for detection of breathing signals.	Recruiting, no results posted.
The role of dietary calcium for treatment of osteopenia in girls with RTT NCT05352373 [96]	Not provided	28 April 2022	Baylor College of Medicine	Randomized, placebo-controlled trial of oral calcium supplementation for osteopenia in girls and women with RTT. The goal is to increase body bone mineral content and density.	No results posted.
Creatine metabolism in RTTNCT01198015 [97]	Maastricht, Belgium	17 February 2011	Maastricht University Medical Center	Many RTT patients meet criteria for moderate to severe malnutrition.	No results posted.
Phase 2 study of EPI-743 for treatment of RTT NCT01822249 [98]	Siena, Italy	26 July 2018	Edison Pharmaceuticals Inc.	EPI-743, structurally related to vitamin E, has demonstrated efficacy and safety in treatment of disorders with oxidative stress. This is a phase 2 study to evaluate treatment of Rett with Epi-743.	Those treated with EPI-743 showed significant increase in head circumference; improvements in oxygenation, hand function and disease biomarkers were observed.
Development of a behavioral outcome measure of Rett syndrome (RettBe) NCT03196323 [99]	Tampa, United States	9 December 2020	University of South Florida	Developing a prototype for a behavioral questionnaire to standardize quantifications of behavioral outcomes in treatment trials and clinical practice.	No results posted.
‘Uptime’ participation intervention in girls and women with RTT NCT03848442 [100]	Copenhagen, Denmark	21 February 2019	Rigshospitalet, Denmark	Aim of study is to evaluate feasibility and health-related improvements of individualized 12 week ‘Uptime’ participation to increase physical activity in girls and women with RTT.	The Uptime intervention was perceived as feasible by caregivers. Small to medium effects were seen in decreasing sedentary time, increasing daily step count, walking capacity and quality of life. Positive effects were decreased sedentary time and increased walking capacity at short-term follow-up.
Effects of creatine supplementation in RTT NCT01147575 [101]	Vienna, Austria	22 June 2010	Medical University of Vienna	Randomized double-blind controlled trial of creatine supplementation to investigate whether changes in biochemical or clinical parameters can be observed through increased availability of labile methyl groups for different methylation reactions.	Creatine monohydrate supplementation increased global DNA methylation, but scores reflecting clinical improvement were lower for creatine than for placebo.
A study to evaluate ketamine for the treatment of RTT NCT03633058 [102]	United States: Alabama, Colorado, Illinois, Massachusetts, Pennsylvania, Tennessee, Texas	22 April 2022	Rett Syndrome Research Trust	Phase 2 clinical trial assessing oral ketamine for treatment of RTT. Efficacy assessed through physician and caregiver questionnaires and continuous, wearable, at-home biosensor data collection.	Data being evaluated to determine support for potential phase 3 trial. Food and Drug Administration has granted orphan drug designation to racemic ketamine (Ketarx).
New genes involved in molecular etiology of RTT through DNA microarray comparative genomic hybridization NCT02885090 [103]	France	31 August 2016	Central Hospital, Nancy, France	Searching for pathogenic chromosomal imbalance through comparative genomic hybridization on DNA microarrays on typical and atypical RTT. Bioinformatics approach will look for candidate genes confirmed by classical mutation screening in RTT.	No results posted.
Pilot study of effects of desipramine on neurovegetative parameters of the child with RTT NCT00990691 [104]	Marseille, France	26 July 2018	Assistance Publique Hopitaux De Marseille	In mouse models, it has been shown that endogenous noradrenaline helps to maintain normal respiratory rhythm. Desipramine, a selective inhibitor of norepinephrine reuptake, is efficient to reduce respiratory alterations in *Mecp2*-deficient mice. Phase 2 clinical trial to study efficacy and safety of desipramine in children with RTT.	No significant difference between groups with high and low desipramine dose, no clinical efficacy shown. However, it shows additional reasons to test noradrenergic pathway in RTT due to correlation between desipramine concentration and apnea–hypopnea index.

Summarized clinical trials. CSF is cerebrospinal fluid, OMT is osteopathic manipulative treatment, and DNA is deoxyribonucleic acid.

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
