# Peer review of "Rett Syndrome and the Role of MECP2: Signaling to Clinical Trials"

_brainsci, 2024, doi:10.3390/brainsci14020120_

Round 1

Reviewer 1 Report

Comments and Suggestions for Authors

The manuscript titled “RETT SYNDROME AND THE ROLE OF MECP2: SIGNALING TO CLINICAL TRIALS” authored by Adele Lopes and collaborators summarizes recent advancements in Rett syndrome and therapeutic approaches focused on MeCP2. The paper is well-written, well-structured, and provides a comprehensive overview.

One suggestion for improvement would be to include a brief paragraph within Chapter 2: MeCP2 FUNCTION, discussing the modulations observed in metabolism, specifically highlighting cholesterol metabolism in patient-derived cells and plasma/serum. This addition could you provide some insights on  the interplay between MeCP2 and metabolic pathways in Rett syndrome.

Author Response

Thank you very much for taking the time to review this manuscript and valuable suggestions. Please find the responses below and corrections highlighted in the re-submitted files.

Reviewer Evaluation

1) The manuscript titled “RETT SYNDROME AND THE ROLE OF MECP2: SIGNALING TO CLINICAL TRIALS” authored by Adele Lopes and collaborators summarizes recent advancements in Rett syndrome and therapeutic approaches focused on MeCP2. The paper is well-written, well-structured, and provides a comprehensive overview.

            One suggestion for improvement would be to include a brief paragraph within Chapter 2: MeCP2 FUNCTION, discussing the modulations observed in metabolism, specifically highlighting cholesterol metabolism in patient-derived cells and plasma/serum. This addition could you provide some insights on the interplay between MeCP2 and metabolic pathways in Rett syndrome.

            We would like to thank the reviewer for his excellent feedback. Indeed, adding details about cholesterol metabolism improves the manuscript and we have addressed the reviewer suggestion by adding an extra paragraph on how the loss of MECP2 has shown to modulate the cholesterol homeostasis (pages 3 & 4, lines 88-115). The changes are highlighted in the manuscript.

Reviewer 2 Report

Comments and Suggestions for Authors

The narrative review focuses on Rett syndrome a well-known diagnostic entity. The authors have identified the main areas that they need to cover in the review and include most of the essence. In writing a review on a well-established topic, a narrative review should focus on recent advances in the topic including as many papers from the last three years as possible. The main area for improvement is to re-organize the content under each subheading. Identify what content is to be outlined under each sub-title and organize them nicely. 

Comment 1

Follow the HGNC guidelines for Human Gene Nomenclature

Bruford EA, Braschi B, Denny P, Jones TEM, Seal RL, Tweedie S. Guidelines for human gene nomenclature. Nat Genet. 2020 Aug;52(8):754-758. doi: 10.1038/s41588-020-0669-3. PMID: 32747822; PMCID: PMC7494048.).

Authors have used MECP2, MeCP2, and Mecp2. The HGNC Approved Gene Symbol: MECP2

Line 256; “four IEGS, fos, junb, egr2, and npas4” are also in small letters and non-italics.

Use the correct symbol and be consistent. genes names are italicized while protein names are not. Consistently use the name in the text

Comment 2

Referencing style

Use the MDPI referencing style. For instance;

Line 126; common in X-linked neurological disorders (26) (27).

Line 136; clinical presentations (29) (30).

These should be corrected as [26,27] and [29,30).

Comment 3

Reorganize the introduction.

The introduction should ideally give an overview of the general facts of Retr syndrome before the reader moves on to subsequent sections for more details.

What is Rett syndrome; The main symptoms (not included), gene involved, inheritance (not included), prevalence, female preponderance (not included), first reported case, and timeline. prognosis/outcome (not included), or any treatment/supportive care (not included). 

Pathogenic variants of MECP2 can cause several variants of Rett Syndrome and phenotypes other than Rett Syndrome Please consult MIM  * 300005, and outline these briefly in the introduction.

In the Figure 1 legend include citations for the references you used to make the timeline. Otherwise its plagiarism.

The last paragraph of the introduction includes information that needs to be included in subsequent sections (e.g. MeCP2 MUTATIONS & MeCP2 FUNCTION).

Comment 4

Reorganize the section “MECP2 Functions”

“MeCP2 mutations are not only found in girls but also have lately been found in boys: producing atypical RTT phenotype and other less distinctive RTT features that lead to a new diagnosis of male RTT encephalopathy.” I think this sentence stands alone in the section for “MECP2 Functions” and should be shifted to the introduction and/or “MeCP2 MUTATIONS”.

The last paragraph  (Because of the similarity between autism and Rett syndrome……) is not relevant to the section “MECP2 Functions”. Move this section to a relevant section.

Comment 5

Reorganize the section “MECP2Mutations”

ACMG recommends that the terms mutation and polymorphism be replaced by the term “variant” with the following modifiers: (1) pathogenic, (2) likely pathogenic, (3) uncertain significance, (4) likely benign, or (5) benign as the terms “mutation” and “polymorphism”, which have been used widely, often lead to confusion due to incorrect assumptions of pathogenic and benign effects respectively.

I would change the title to “MECP2 variants” and replace the term mutation in the text as appropriate.

Reorganize the whole section.

First outline the mutation spectrum indication type of mutations, location, origin (de-novo?), reported countries, common mutations, patterns of distribution across words, and pathogenicity. Then finally genotype-phenotype correlations.

Comment 6

Better avoid the abbreviation RTT in the sub-titles. E.g. “STAGES OF RTT AND SYMPTOMS”

Comment 7

Figure 4; the font is blurred, with poor resolution.

Comment 8

What is the purpose of including two tables? Content in each table is different?

in the table legends, define the abbreviations used in the table contents (e.g. CPK, BDNF, IGF-1, etc)

All the best

Comments on the Quality of English Language

 language is fine, only minor editing required

Author Response

Thank you very much for taking the time to review this manuscript. The reviewer’s detailed feedback and suggestions are very valuable to improve the manuscript and very much appreciated. Please find the detailed responses below and corrections highlighted in the re-submitted files.

The narrative review focuses on Rett syndrome a well-known diagnostic entity. The authors have identified the main areas that they need to cover in the review and include most of the essence. In writing a review on a well-established topic, a narrative review should focus on recent advances in the topic including as many papers from the last three years as possible. The main area for improvement is to re-organize the content under each subheading. Identify what content is to be outlined under each sub-title and organize them nicely. 

We thank the reviewer for the overall suggestion, and we have added now several paragraphs highlighting important updates and promising basic and clinical research findings in the last 3-4 years. In addition, we have changed the titles and subtitles as suggested by the reviewer. However, we did not add more subtitles as this is a concise review and hence, we did not add subtitles for smaller topics, rather we organized it under a more broader title. With the new title changes and addition of several paragraphs of updated content, we hope the manuscript is more refined and we thank the reviewer and appreciate his/her feedback.

1) Follow the HGNC guidelines for Human Gene Nomenclature. Use the correct symbol and be consistent. genes names are italicized while protein names are not. Consistently use the name in the text

            Thank you for your comment. We have consistently renamed gene names based on the gene nomenclature. For example- changed MECP2, FOS, JUNB, EGR2, and NPAS4 correctly according to the correct symbol. We have high-lighted these changes in the manuscript as well. Whenever mouse genes are mentioned, we used lower case letters. `

2) Referencing style. Use the MDPI referencing style.

            Thank you for this feedback. We have re-formatted the manuscript references according to the MDPI Brain Sciences format.

3) Reorganize the introduction.

            The manuscript introduction has been re-organized in a few places and citations for first figure has been included according to the reviewer’s suggestions. The last paragraph of introduction is about orienting readers regarding the areas we will be touching base further in the manuscript and we explain in detail in the following paragraphs under appropriate sections. However, we did not add all the details requested by the reviewer in the introduction as the other reviewer requested us to cut down the introduction further as the history and Rett syndrome background information is covered in several other reviews in the field and he/she would prefer this review to be an update in the field for the last 3-4 years. We did not completely remove the introduction but reorganized a bit according to this reviewer’s suggestion and strived to strike a fine balance between both reviewer’s comments. Our intention is to make this manuscript a standalone article for readers outside of this field and at the same time serve as an update for readers in neurodevelopmental disorder field. It is a fine line and an impossible feat to achieve but we tried our best. So, we gave a few lines about history and MECP2 mutations to orient readers about Rett syndrome and then go into detail about signaling and promising clinical research etc.

4) Reorganize the section “MECP2 Functions”

 We thank the reviewer for the suggestions. We have moved the appropriate sentences and paragraphs as suggested by the reviewer and highlighted accordingly.

5) Reorganize the section “MECP2Mutations”

We thank the reviewer for the suggestions. We have edited the appropriate sections and paragraphs as suggested by the reviewer and highlighted accordingly. We also renamed the title as suggested by the reviewer.

6) Better avoid the abbreviation RTT in the sub-titles. E.g. “STAGES OF RTT AND SYMPTOMS”

The suggested change has been implanted accordingly.

7) Figure 4; the font is blurred, with poor resolution.

 We apologize for the tardiness in the figure. We have replaced with a high-resolution picture and if requested by the journal, we can also provide the figure in other high-resolution formats.

8) What is the purpose of including two tables? Content in each table is different? in the table legends, define the abbreviations used in the table contents (e.g. CPK, BDNF, IGF-1, etc)

Table 1 is completed trials and other table contains trials not linked in gov trials website. When the formatting to brain sci template was done, the legend was truncated somehow, and we apologize for the confusion. This has been fixed now and we also added the abbreviations as per the reviewer suggestion.

Reviewer 3 Report

Comments and Suggestions for Authors

To the Authors

In the present Ms., the AA are reviewing the issue of RTT and MECP2-signaling and on the lessons learned from clinical trials. Overall the Ms is well written and planned. Language is appropriate. However, I see a large clear focus on very mainstream information rather than an attempt to revise the upcoming issues concerning this quite paradigmatic, rare, genetically-determined neurodevelopmental disorder affecting almost exclusively the female gender.  

Suggestions

1.    The background information /history of RTT discovery is very well known and treated in several excellent publications. I would suggest to summarize this part into a more concise and up-to-the-point one.

2.    In my opinion, the review focuses on very well-known mainstream trials while it very little o no attention to promising lessons learned from experimental work in gut-brain-axis and oxidative stress. I would suggest to pay more attention to the emerging fields in RTT experimental and clinical research in order to give to the general readership, as well as to researchers in the field, a more comprehensive and balanced view on this syndrome.

Author Response

We would like to thank the reviewer for his/her valuable time in reviewing the manuscript and giving feedback and suggestions. Please find the responses below and corrections highlighted in the re-submitted files.

Reviewer Evaluation

  1. In the present Ms., the AA are reviewing the issue of RTT and MECP2-signaling and on the lessons learned from clinical trials. Overall the Ms is well written and planned. Language is appropriate. However, I see a large clear focus on very mainstream information rather than an attempt to revise the upcoming issues concerning this quite paradigmatic, rare, genetically-determined neurodevelopmental disorder affecting almost exclusively the female gender.  

 The background information /history of RTT discovery is very well known and treated in several excellent publications. I would suggest to summarize this part into a more concise and up-to-the-point one.

We thank the reviewer for this suggestion. We have only one 6-10 lines about the history and background and rest of the progression is made into Figure 1 in a concise manner. While we understand the reviewer’s point of view, we would also like our article as a stand-alone reading manuscript for readers who are from outside the field while simultaneously serving as an update for readers in the neurodevelopmental disorder field. We understand it is a hard balance to achieve but we tried our best. That is the reason we have figure 1 to give a snapshot of the history in the Rett syndrome field and to our knowledge there are only a few figures like this. We hope the reviewer can appreciate our fine balance between not too much history but also just enough for a reader from outside field to appreciate the overall perspective of this rare disorder.

  1. In my opinion, the review focuses on very well-known mainstream trials while it very little o no attention to promising lessons learned from experimental work in gut-brain-axis and oxidative stress. I would suggest to pay more attention to the emerging fields in RTT experimental and clinical research in order to give to the general readership, as well as to researchers in the field, a more comprehensive and balanced view on this syndrome.

            We thank the reviewer for this excellent suggestion and indeed, we have now added multiple recent promising experimental and clinical research including gut-brain axis and oxidative stress. We have added a big paragraph on page 8 (lines 308-335) and another paragraph on page 14 and updated the corresponding references accordingly.

Round 2

Reviewer 2 Report

Comments and Suggestions for Authors

Dear Authors,

Many thanks for addressing my comments and sending a revised with changes highlighted and a point-by-point response. To improve your manuscript further I would recommend citing primary references as much as possible. I don't think it's advisable to publish your paper without fixing this. 

Regarding the references that you newly included in the figure legends (according to my suggestions)

- many are not primary references (it's also a form of plagiarism as we are not giving due credit to the original scientists who played a vital role in milestones in the identification of advances in Rett Syndrome.

e.g. you have not cited the clinical trial in the legend of Figure 1. 

Another example; Line 42-43; "Bienvenu et al. is exon 3 which includes the methyl binding 42 domain and the transcriptional repression domain of the gene [3]." 

- you are giving credit to Bienvenu et al. in the sentence, without citing the primary resource. what you have cited is a review by Liyanage et al.

-Likewise, Lines 96-104. Buchovecky et al ? You have not cited his work but of Segatto 

There are several other examples where the primary source is not cited (but a secondary source is cited. 

I hope you will go through the whole paper carefully and revise the references before publication. 

All the best

Author Response

Thank you very much for taking your time and letting us know about the references. We have completely checked and revised our references by citing the primary references. We apologize for the error. 

Reviewer 3 Report

Comments and Suggestions for Authors

I thank the AA for their efforts in addressing my two main points.

Point 1) More concise background information: while I am not encouraging a common trend to repeat very well know information, I clearly understand the AA point of view and I appreciate their efforts in justifying the need of explaining to background information for an out-of-the-field readership

Point 2) Attention to promising lessons learned from experimental work in gut-brain-axis and oxidative stress: I do appreciate the AA for their efforts in satisfactorily addressing this point.

Author Response

Thank you very much for taking the time to review this manuscript.